# Contributions of Microelectrochemical Scanning Techniques for the Efficient Detection of Localized Corrosion Processes at the Cut Edges of Polymer-Coated Galvanized Steel

**DOI:** 10.3390/molecules27072167

**Published:** 2022-03-27

**Authors:** Dániel Filotás, Javier Izquierdo, Bibiana M. Fernández-Pérez, Lívia Nagy, Géza Nagy, Ricardo M. Souto

**Affiliations:** 1Department of General and Physical Chemistry, Faculty of Sciences, University of Pécs, Ifjúság útja 6, 7624 Pécs, Hungary; lnagy@ttk.pte.hu (L.N.); g-nagy@gamma.ttk.pte.hu (G.N.); 2János Szentágothai Research Center, University of Pécs, Ifjúság u. 20, 7624 Pécs, Hungary; 3Department of Chemistry, Universidad de La Laguna, P.O. Box 456, 38200 La Laguna, Tenerife, Spain; jizquier@ull.edu.es (J.I.); bfernand@ull.edu.es (B.M.F.-P.); 4Institute of Material Science and Nanotechnology, Universidad de La Laguna, 38200 La Laguna, Tenerife, Spain

**Keywords:** localised corrosion, coatings, cut edge, galvanized steel, scanning vibrating electrode technique, scanning electrochemical microscopy, micro-potentiometry, ion-selective microelectrodes, galvanic couple

## Abstract

Spatially resolved information on corrosion reactions operating at the cut edges of coated metals can be obtained using microelectrochemical scanning techniques using a suitable selection of operation modes and scanning probes. The scanning vibrating electrode technique (SVET) provides current density maps with a spatial resolution of the order of the dimensions of the sample, which allows the temporal evolution of the corrosion reactions to be followed over time. This leads to the identification and localization of cathodic and anodic sites, although the technique lacks chemical specificity for the unequivocal identification of the reactive species. The application of scanning electrochemical microscopy (SECM) was previously limited to image cathodic reaction sites, either due to oxygen consumption in the amperometric operation or by the alkalinisation of the electrolyte in potentiometric operation. However, it is shown that anodic sites can be effectively monitored using an ion-selective microelectrode (ISME) as a probe. The ISME probes detected differences in the local concentrations of Zn^2+^ and OH^−^ ions from the cut edges of a complete coil coating system compared to the same system after the polymeric layers were removed. In this way, it has been shown that the inhibitor loading in the polymer layers effectively contributes to reducing the corrosion rates at the cut edge, thus helping to extend the useful life of the sacrificial galvanized layer bonded directly to the steel matrix. Additionally, these two probe configurations can be integrated into a multi-electrode tip for potentiometric operation to simultaneously monitor localized changes in pH values and metal ion dissolution in a single scan. Spatial and temporal distributions were further investigated using different rastering procedures, and the potential of constructing pseudomaps for 2D-imaging is described.

## 1. Introduction

Polymer coated galvanized steels find widespread technological application when lightweight structures are required. Efficient corrosion resistance is provided to carbon steel matrices by the galvanic pair effect produced by a thin layer of zinc or its alloys. Additional protection to the metallic coating is usually attained by the application of an organic coating to the galvanized layer that acts as a physical barrier against water and ions accessing the metal surface, preventing the onset of electrochemical reactions there. Although such composite materials quite efficiently protect the underlying metal matrix when defects are produced by the action of the environment (scratches, particle impacts, etc.), greater corrosion activities occur at the cut edges of the coated metal, and the sacrificial protection mechanism provided by the less noble metals in the galvanized layer must operate with an adversely small area ratio between the thickness of the metallic coating and the cross section of the bulk metal to protect. In addition, the electrochemically driven corrosion processes occurring at cut edges are highly localized and heterogeneously distributed throughout the system. Thus, there is need for experimental techniques with spatial resolution that acquire data in real time in order to provide corroborative or other novel evidence of the underlying reaction schemes and the effectiveness of anticorrosion protection formulations, which are typically based on the analysis of electrochemical impedance (EIS) data, although these are surface averaging measurements [1,2,3].

The option of scanning microelectrochemical techniques is greatly contributing to the knowledge and analytical monitoring of corrosion reactions at the microscale. Three main groups of scanning microelectrochemical techniques have been introduced to the investigation of degradation reactions at coated metals, which can be classified as those based on the measurement of voltage fields in the electrolyte using a reference microelectrode as scanning probe (i.e., the Scanning Reference Electrode Technique (SRET) [4] and the Scanning Vibrating Electrode Technique (SVET) [4,5]), the measurement of faradaic currents due to some redox conversion at the probe or tip (namely, the Scanning ElectroChemical Microscope (SECM)) [6,7,8], or the measurement of local electrochemical impedances (using the Localized Electrochemical Impedance Spectroscopy (LEIS)) [9]. In the first group of techniques, the sensing probe is a reference microelectrode that is scanned over the sample in either a static motion in SRET or subjected to a controlled vibration for SVET, whereas another microelectrode is set stationary in the solution as reference [5]. In this way, potential gradients in the electrolyte adjacent to the sample caused by local ionic currents occurring around anodic and cathodic areas can be detected with respect to values obtained in the bulk solution away from the sample. The resource to probe vibration enables greater potential resolution in the system [5]. The resulting electrochemical image contains either the potential distribution or the local current density values in the proximity of the scanned surface. Until now, the most successful application of a scanning microelectrochemical technique to the investigation has been achieved using SVET [10,11,12,13,14,15,16,17,18,19,20,21,22,23]. In addition to monitoring the different extent of the localized corrosion reactions occurring at the cut edges due to different compositions of the coating layers applied on the base metal, the occurrence of local pH changes has been also shown using ion-selective electrodes, either in a separate experiment [15] or quasi-simultaneously in the case of assembling together the two different probe holders [24]. pH changes are the result of both the hydrolysis of the dissolving metal arising from the anodic sites [25], as well as the cathodic consumption of either oxygen in neutral and alkaline environments, or hydrogen ions in an acid medium [26]. Thus, local pH changes are important traits of a corroding system. Unfortunately, the local potential and ionic current fluxes data obtained using SRET and SVET, as well as for impedance measurements using LEIS, lack chemical selectivity and thus can only detect the sites on the sample associated with either cathodic or anodic half-cell reactions, but cannot serve to identify the chemical species involved in those reactions.

Alternately, an active microelectrode is employed in SECM, thus the electrochemical signal recorded at the tip is actually the current flowing through it as result of electron transfer processes on its active surface. Therefore, redox conversion of a suitable species present in the solution must occur, utilizing the advantage that the measured current is highly sensitive to both the tip-sample distance and the electrochemical activity of the sample, whereas the tip is operated amperometrically. The redox species employed for monitoring can be either added to the measuring system for chemical selectivity (i.e., feedback operation modes [7]), or to be generated in the surface as result of the degradation processes (sample generation—tip collection SG/TC mode [7]), or naturally present in the environment yet consumed in the degradation reaction of the surface (redox competition mode [27,28,29]). In every case, the recorded current flowing at the tip contains both topographical and chemical information on the system, and can be correlated to a certain chemical species by adequately tuning the potential applied to the tip. The applicability of this procedure was already tested by Souto and coworkers, although SECM in amperometric operation could only be used to monitor the oxygen consumption related to the cathodic process, whereas the actual metal dissolution sites could not be monitored directly due to the unavailability of further oxidized states in the redox conversion of dissolved Zn(II) ions at the tip [30]. As result, the SECM were usually considered in combination with either ionic current measurements using SVET [20] or local pH monitoring using a pH-sensitive probe [31,32]. In all these cases, measurements could not be performed simultaneously as they either required using a separate probe holder [33] or instrumental set-up [23], or the exchange of the measuring probe [32].

An alternate route to achieve simultaneous monitoring of the cathodic and anodic processes at cut edges can be achieved by using ion-selective microelectrodes (ISME) as the probe [34]. In the case of galvanized steel, the use of a Zn^2+^ ion-selective microelectrode (ISME) [35] would be an obvious choice in addition to pH monitoring. This ISME selectively reaches an equilibrium Donnan potential that varies logarithmically with Zn^2+^ ion concentration as described elsewhere [26]. Another development will be the combined use of multiple potentiometric sensors in a single probe through the use of multibarrel electrode configurations. That is, by building a single probe assembly containing two or more microelectrodes for selective chemical sensing with spatial resolution [36,37,38,39]. In particular, combined distributions of Zn^2+^ ions and pH can be performed with the same probe in a single scan. The proof of concept was demonstrated by imaging model galvanic couples of dissimilar metals of similar surface areas, but it will be further developed in this contribution for the monitoring of a cut edge system presenting a big difference in size between the anodic and cathodic regions due to the small Zn:Fe ratio in the investigated surface. Additionally, procedures to obtain chemically resolved images of the complete cut-edge exposed to the aggressive environment have been explored, allowing us to overcome the previous limitation of SECM investigations that were based solely on a small selection of scan lines due to slower acquisition times compared to SVET or LEIS. 

## 2. Materials and Methods

### 2.1. Materials

Cut edges of coil-coated galvanized mild steel were investigated. The mild steel foil was 400 μm thick, whereas the galvanized layers applied at both sides were 30–35 μm thick and coated with a 5 μm polyester primer containing strontium chromate as inhibitor. One of the sides was further coated with a 20 μm white polyester topcoat containing TiO_2_. Additional details on the preparation of the coil coating system and its adhesion characteristics were reported elsewhere [40,41]. This system was embedded in epoxy resin. The resin and the hardener were mixed in an 8:1 ratio in a homemade mold, and allowed to cure for 12 h at room temperature. Afterwards, the exposed surface was wet ground with a sequence of 220–4000 grit SiC papers under ethanol, followed by polishing with 0.5 μm alumina slurries. The complete cut edge system is shown in Figure 1A,B. An additional sample was prepared by removing the paint layers, and the resulting cut edges are sketched in Figure 1C.

### 2.2. Scanning Microelectrochemical Measurements

In the scanning vibrating electrode technique, the detection probe is a single 10 μm diameter PtIr wire with an electrochemically grown platinum black deposit to achieve sufficient capacitance (66.7 µF for the probe used to record the SVET data shown below). The detection probe vibrates at low amplitude thanks to piezoelectric holders driven by sinusoidal oscillators at two different optimized frequencies, one for each axis of vibration. In this work, probe vibrations of 20 μm amplitude were applied in the directions normal and parallel to the surface, with respective vibration frequencies of 78 and 185 Hz, and maintaining a probe–substrate distance of 100 μm. The positioning distance of the tip to the sample was performed avoiding tip crushing using a video camera connected to an optical microscope, and this system was also used to track the movement of the vibrating microelectrode on the sample during the operation. The electrochemical cell contains another microelectrode set stationary in the solution as a reference, and the sample placed facing up in its bottom. The SVET equipment was supplied by Applicable Electronics Inc. (Forestdale, MA, USA), and used with a personal computer running ASET 2.00 software.

SECM experiments were carried out using an instrument manufactured by Sensolytics (Bochum, Germany) and operating with an Autolab bipotentiostat equipped with a frequency response analyzer, all controlled by a personal computer. A homemade voltage follower based on an operational amplifier with an input impedance of 10^12^ Ω was interconnected between the cell and the potentiometric input of the system. SECM measurements were performed on cut edge samples immersed in 1 mM NaCl solution. There was a camera microscope inside the Faraday cage, which helped in the positioning of the tip electrode by the gentle approach procedure. The surface of the mild steel looked like a mirror after careful polishing, and the microelectrode tip was positioned on the sample in the central area and brought close to the surface. Once the electrode abuts the reflection of the tip, the *Z* motor was lifted 20 μm. The scan rate was 10 μm s^−1^, and the scan length was 2000 μm from the left to the right in a direction perpendicular to the longest axis of the cut edge.

Single barrel antimony microelectrodes have been fabricated for the measurement of local pH distributions in the electrolyte adjacent to the cut edge systems under investigation. Their structure is shown in Figure 2A. First, a few grams of antimony powder were melted and drawn under suction into a thick-walled capillary using a large syringe (50 mL). Using metal tweezers and a roaring flame from the Bunsen burner, smaller capillaries filled with antimony were pulled. The small capillaries were checked under an optical microscope and the parts without bubbles were chosen. One of these parts was inserted and attached to the tip end of the microelectrode body. The electrical contact between a recently abraded copper wire and the antimony fiber was made with liquid mercury. The diameters of the antimony fibers used in the experiments were between 15 and 30 µm. The calibration procedure for the pH-sensitive antimony electrode was as follows: the electrodes were then exposed to calibration buffers with a pH range of 11 to 4, respectively. Sometimes it was necessary to renew the Sb_2_O_3_ layer on the surface of the antimony electrode. In these cases, the electrodes were soaked in the pH = 4 buffer solution for a few minutes. Alternatively, renewal of the surface layer could be achieved by biasing at 0.0 V vs. Ag/AgCl/KCl (3 M) for 1 min. A typical calibration curve can be seen in Figure 2B, which was recorded in eight standard solutions and monitored versus the Ag/AgCl/KCl (3 M) reference electrode.

The preparation and calibration of a single barrel Zn^2+^ ion-selective microelectrode with internal solid contact are described next. Zn-ISME tips were fabricated from borosilicate glass capillaries using a Model P-30 puller. The silanization method of the tips was as follows: the tip was soaked in a 5% solution of dichloro–dimethyl–silane in CCl_4_, then placed in a closed Petri dish and placed in an oven at 120 °C for 30 min. The preparation of the solid contact Zn-ISME was continued with the electrochemical polymerization of PEDOT on a 33 µm diameter carbon fiber. A carbon fiber about 1–2 cm long was chosen and attached to a copper wire with silver epoxy adhesive. The wire was used as the working electrode in an ionic liquid solution of BMIM^+^ PF_6_^−^ containing 0.1 M EDOT, and the electrochemical cell was completed with a silver reference and a platinum auxiliary electrode. In the polymerization stage, 10 consecutive cyclic voltammetric cycles were performed in the range −0.9 ≤ *E* ≤ 1.3 V. The PEDOT-coated carbon fiber was introduced into a 0.1 M KCl aqueous solution and subjected to 15 consecutive cycles in the range −0.9 ≤ *E* ≤ 0.8 V, with a scan rate of 0.05 V s^−1^, in order to dope the polymer coating. Each coating was tested by cyclic voltammetry in the same solution, in the range −0.4 ≤ *E* ≤ 0.5 at 0.05 V s^−1^; five consecutive scans were performed. A small amount of the ion-selective cocktail was placed into the tip of the micropipette using a suitable syringe. The cocktail contained 98 mg tetrahydrofuran, 42 µL emollient 2–nitrophenyl–octyl ether, 2.26 mg PVC, 0.99 mg ionophore, and 0.22 mg potassium–[tetrakis–4–chlorophenyl]–borate. The evaporation of THF in the capillary took about one day. The 33 µm diameter PEDOT-coated carbon fiber attached to a copper wire with silver epoxy was inserted into the capillary lumen, such that the end of the carbon fiber was immersed in the ion-selective cocktail as close to the tip as possible. The electrical lead wire was secured with an instant adhesive, and the electrode was stored for approximately 24–36 h to allow the THF to evaporate. Figure 3A shows a sketch of the single barrel Zn-ISME. The calibration of the Zn-ISME was performed by introducing it into ZnSO_4_ solutions in the concentration range 10^−1^–10^−6^ M. The process started with the most diluted solution and then subsequently introduced the solutions of increasing concentration. After a few seconds, the potential reaches a stable value. The activity coefficients are not negligible in the case of 2:2 electrolytes; they were calculated using the Debye–Hückel theory. The calibration plot of the solid contact ISME can be seen in Figure 3B. In a previous work [42], the pH dependence of the potential of this Zn^2+^-ISME was shown to be negligible for pH ≤ 9.

In addition, double barrel Zn ISME/Sb electrodes were fabricated for simultaneous measurements of local chemical changes in a self-corroding system. They were constructed using the design shown in Figure 4. The bodies of these probes were made from borosilicate capillaries and they were pulled with the Narishige PE-2 instrument. The first step was to join together the two capillaries using a copper wire. Under heating, the capillaries softened and were twisted 180 degrees. After cooling, the double barrel capillaries were pulled under another heating step. They were silanized by soaking in 5% dichloro-dimethyl-silane in CCl_4_ solution, then dried in an oven at 120 °C for 30 min.

The preparation of the Zn-ISME involved the electrochemical polymerization of poly-(3,4-ethylenedioxythiophene) (PEDOT) on a 33 μm diameter carbon fiber [34]. A small amount of Zn-ion selective cocktail was allocated inside the micropipette tip, and the THF solvent was allowed to evaporate for approximately 24 h before the PEDOT-coated carbon fiber was inserted into the capillary lumen. In this way, the end of the carbon fiber was submerged in the ion-selective cocktail. The pH measurements were made using an antimony microelectrode constructed by melting antimony powder inside the corresponding borosilicate capillary [43]. Electrical contact with the resulting antimony fiber was achieved by introducing a copper wire and liquid mercury. Calibration of the antimony and the Zn-ion selective microelectrodes of the double-barrel assembly was performed as described for the corresponding single-barrel ISME using a set of concentration-controlled solutions encompassing suitable pH and/or Zn(II)-ion concentrations. In the case of the double barrel assembly described here, the corresponding calibration curve for the antimony electrode as a function of pH is shown in Figure 5A for 4 ≤ pH ≤ 11, while that of the Zn-ISEM shown in Figure 5B was obtained in the range 1.5 ≤ pZn ≤ 5. The slopes of the calibration curves for the two microelectrodes in the double barrel assembly are very close to those already reported for the corresponding single barrel microelectrodes [43,44].

## 3. Results and Discussion

Corrosion activity was monitored over the cut edge systems exposed to 1 mM NaCl solution. Two different cut edge samples, respectively with and without polymer coating layers applied on the opposite sides of the galvanized layers in contact with the carbon steel sheet (see Figure 1C), were tested to verify the possibilities and limitations of SVET and SECM-based electrochemical imaging proposed in this work. The resulting spatially resolved distributions of the electrochemical activity of the corrosion systems were recorded by moving the corresponding detection probe parallel to the surface to obtain line scans and 2D-maps of the cut edge at different times of immersion in the test electrolyte.

### 3.1. Current Density Distributions Measured over the Cut Edge Systems by SVET

Current density maps were measured every 15 min for the cut edge systems exposed to 1 mM NaCl solution for 3 h, with each map requiring approximately 10 min to complete. Figure 6 shows the first and last recorded SVET maps for the cut edge system only with the galvanized zinc layers on both sides of the mild steel matrix (i.e., the sample configuration corresponding to sketch #2 in Figure 1C). The current density values were compared to those measured in the bulk of the solution. In the vicinity of the resin there was not significant difference with the reference in the bulk of the electrolyte. However, negative and positive deviations, indicated by blue and red coloration, respectively, could be observed over the cut edge sample showing cathodic and anodic regions developed on the surface. The very small surface area ratio of zinc to iron provided poor protection against corrosion of the cut edge.

Cathodic and anodic activities increased with the elapse of time for this system, as can be seen in Figure 7, which represents the time courses of measured current densities at the main peaks associated with adjacent cathodic (blue) and anodic (yellow and red) regions in Figure 6B. The correlation is not linear between the two trends, although they move similarly at the same times. This character of anodic and cathodic locations close together at the cut edge is observable at all times (Appendix A shows the full set of measured SVET maps for this system #2). Note that despite the very small zinc–iron area ratios that occur at the cut edge, the anodic and cathodic sites are well resolved in the SVET maps. Another feature is that the location of the anodic sites occurs preferentially in one of the two galvanized layers (corresponding to the left in the SVET images of Figure 6), and the cathodic sites are observed in the adjacent steel matrix on the right, up, and down.

The same procedure was performed with the complete coil coating system, and two selected SVET maps are shown in Figure 8. Significantly lower current density values were observed in this case compared to the sample with only the galvanized coatings. Furthermore, the overall electrochemical activity remained almost constant most of the time, and no signs of additional activation were observed during the duration of the experiments, a significantly different behavior than that shown in Figure 7 by the sample without the organic coating layers. Another big difference is related to the location of the anodic sites, as only anodic regions extending between the two zinc layers on each side of the steel matrix were apparently observed, which were actually merged above the latter due to the divergence of the diffusion lines for zinc(II) ions released from the dissolution sites of a smaller size. That is, there is no measurable difference in anodic behavior between the two zinc layers in this system, as both contribute to the sacrificial protection mechanism of the steel which behaves like a larger cathode. This effect is best noticed at the beginning of the experiment by inspecting the SVET map in Figure 8A, where two smaller anodic sites occur on each of the zinc layers on both sides of the steel matrix in the central part of the image, and they have not been merged yet, while the reverse has already occurred with the anodic site at the bottom of the same map.

The slower corrosion rates of the complete coil coating system can be related to the precipitation of zinc corrosion products that was observed using the video camera system coupled to the SVET equipment as shown in Figure 9, which depicts the time evolution of a white precipitate. Indeed, before and after recording every map, optical micrographs were taken of the samples, and the evolution of the corrosion process was visually observable from those images. Additionally, the current density scans were found to be in good correlation with those micrographs, as the anodic currents flow where the white precipitate originated from the oxidation of zinc appears, and cathodic areas could be found in all cases.

### 3.2. pH Line Scans Measured over the Cut Edge Systems

As corrosion reactions are associated with pH changes as the result of both the hydrolysis of the anodically oxidized metal and the cathodic consumption of dissolved molecular oxygen in a neutral aqueous solution, local pH changes are relevant features related to the advancement of the degradation process, and they can be monitored using a pH-sensitive microelectrode tip [31]. Cut-edges of the coil coating system with and without the organic coating system depicted by the sketches in Figure 1C were embedded in resin and exposed to naturally aerated 1 mM solution. A series of line scans were recorded over the cut edge system in order to monitor the advancement of the corrosion reaction for each system. Figure 10 shows a selection of five line scans recorded above the complete and the non-painted galvanized steel specimens at various immersion times up to 3 h. They were recorded as the tip moved from left to right across the cut edge samples embedded in epoxy resin. The approximate location of the samples and their various components is shown above the corresponding graphics.

From a cursory inspection of Figure 10, the development of an alkaline electrolyte volume adjacent to the steel surface is found for both samples shortly after exposure to the test solution. This corresponds to the area on the surface of the cut edge where the cathodic half-cell reaction has occurred, namely the reduction of oxygen to hydroxide ions. As a result, local alkalinization was observed in the immediate vicinity of the steel surface, indicating that the cut edge was electrochemically active at early times of exposure. Therefore, the low zinc to iron area ratio clearly resulted in limited protection of the cut edge against corrosion. Although this alkalinization above the steel can be observed for the cut edges of the two samples, significant differences can also be observed for the two systems. Thus, the magnitude and time scale for the alkalinization process was different in each case, because different time evolutions have occurred. The distinctive protective effect of the organic coating and the galvanized layers was also distinguished, as the total pH change was less than one unit in the case of the complete coil coating system of Figure 10A, while it amounted to almost 2 pH units for the steel covered only by the galvanized layers. Furthermore, the decrease in pH change over time was also much slower in the latter. As the polymer coating layers cannot maintain the barrier effect at the cut edge, because the metals both in the galvanized layer and in the steel plate are directly exposed to the electrolytic medium, the beneficial protection effect they provide must come from the action of the corrosion-inhibiting species dispersed in the polymer matrix, which are effectively released from the coating to precipitate on the bare metal surfaces. 

Another characteristic common to both samples is that the pH changes along the curves decreased with time, this effect being rather small for the cut edge of the sample without organic coating layers shown in Figure 10B, while that decrease in pH was very abrupt for the complete coil coating system. Although the transport processes of the ionic species generated at the cut edge to the bulk of the solution (i.e., either ion diffusion or convection effects due to the movement of the scanning tip) can contribute to the decrease in the pH values observed with the elapse of time, a more likely explanation is that the corrosion products of the anodic reaction precipitated on the steel surface, thus forming a barrier that separates the metal surface from the aggressive environment.

In addition, the shapes of the pH plots recorded over the cut edges were also different for each system, with a more symmetric distribution in the case of the system without the organic coating layers (see Figure 10B), while they are clearly asymmetric in the case of the complete coil coating system (cf. Figure 10A). Therefore, the microelectrochemical measurements described here contain both spatially and temporally resolved information about degradation processes at metal cut edges. The difference between the left and the right side of the complete coil coating system in Figure 10A was most noticeable at the longer exposition times. The organic coating contained strontium chromate as a corrosion inhibitor, and this effect was also detectable due to the smaller acidification occurring on the coated Zn area in the case of the complete coil coating galvanized steel system.

The anodic reaction in a Zn-Fe galvanic pair is the dissolution of Zn according to: Zn → Zn^2+^ + 2e^−^(1)
whereas the reduction of dissolved oxygen occurs on the surface of iron: O_2_ + 2H_2_O + 4e^−^ → 4OH^−^(2)

Released zinc(II)-ions may undergo hydrolysis producing either the zincyl-ion or insoluble Zn(OH)_2_, depending on the local pH: Zn^2+^ + H_2_O ⇆ Zn(OH)^+^ + H^+^(3)
Zn(OH)^+^ + H_2_O ⇆ Zn(OH)_2_ + H^+^(4)

Zn(OH)_2_ can further form ZnO, or even Zn(OH)_3_^−^ and Zn(OH)_4_^−^ at higher pH values [45]. Therefore, the lower pH values observed for the galvanized steel sample without the organic coating layers will facilitate metal dissolution and the development of increasingly greater corrosion rates with the elapse of time, whereas further activation of the corrosion process is prevented in the complete coil coating system due to the precipitation of zinc corrosion products, as can be observed in the optical micrographs shown in Figure 9. 

### 3.3. pH Quasi-Maps of the Cut Edge Systems

The results presented in the previous section provided a time-resolved characterization of the samples. However, on the surface of a corroding metal, there could be more distributed anodic and cathodic places. Therefore, 2D maps would be more informative than one-dimensional linear scans. Unfortunately, scanning a relatively large cut edge can be time consuming relative to the rate of the corrosion. However, high scan rates will result in poor resolution and image distortion. In this way, imaging of corrosion activity at cut edges using SECM has always been limited to collecting selected line scans as performed above in Section 3.2, and thus the full description of the cut edge system would require the use of SVET despite its lack of chemical resolution [23].

A promising way to overcome these limitations in SECM to image the entire cut edge system in a single experiment can be opened in potentiometric operation by obtaining 2D-pseudo maps. Although still based on the collection of a small number of linear scans as described before, real-time monitoring of the cut edge can be achieved. The procedure consists of scanning the line above the surface and after each scan, moving the tip along the *Y* axis in a step size equal or bigger than 100 µm. That is, the step sizes on the *X* and *Y* axis are significantly different. In this way, a quasi-map is recorded instead of an actual map, which would require a small fraction of the area covered along each line to be scanned in the following line to continuously and unambiguously resolve the chemical activity of the surface. However, we consider that the quasi-maps recorded as described here can provide a good approximation to resolve spatial distributions and local changes in reactive systems such as the cut edges studied in this work. The proof of concept will be next shown using the pH-sensitive probe to record pH quasi-maps of the degradation process occurring at the cut edge of coil coating steel.

The constructed pH quasi-maps of the cut edges of the complete coil-coated galvanized steel (corresponding to sketch #1 in Figure 1C) and uncoated galvanized steel (corresponding to sketch #2 in Figure 1C) are shown in Figure 11A,B, respectively. It should be noted that the highest pH values recorded there effectively extended beyond the linear calibration range of the pH-sensitive microelectrode, a characteristic clearly related to electric field effects occurring when high concentration distributions develop in a system [46]. In any case, it is considered that significant observations can be drawn from the comparison of the two images in Figure 5. The anodic activity of the polymer-coated sample was observed to be smaller, as evidenced by the lower acidification of the environment, although the areas of exposed metals in the cut edge were basically the same for the coated and the non-coated specimens. This is consistent with the fact that the polymer layers contained strontium chromate as corrosion inhibitor for zinc, and that the complete coil coating system with the polymer layers on each side is more corrosion resistant that the steel coated only with the sacrificial galvanized layer.

### 3.4. Measurements over the Cut Edge Systems Using a Zn ISME

The use of a Zn^2+^-ion selective electrode to measure local distributions of Zn^2+^ ion concentration over the cut edge systems was next investigated by recording line scans over the samples immersed in a 1 mM NaCl solution, using a tip-sample distance of 20 µm. Figure 12 shows the results of a 2 h experiment. The higher electrochemical reactivity of uncoated galvanized steel explains the higher dissolution rates of zinc recorded in Figure 12B. In fact, the pZn values in the line scan recorded after only 13 min of exposure to the test solution in the case of this system are lower than those recorded at any time in Figure 12A for the complete coil coating system. The onset of corrosion product precipitation would explain the almost time-independent curves found in Figure 12A for times greater than 26 min. The precipitates would partially block the metal surface, making it difficult for molecular oxygen to access the electrons released by the metal during ionization. 

### 3.5. Simultaneous Measurements with the Zn ISME-Sb Double-Barrel Microelectrode

Line scans were taken above the complete coil coating sample (corresponding to sketch #1 in Figure 1C) with the Zn-Sb double-barrel electrode using the same conditions as in the single barrel electrodes described above. For comparison purposes, the results of selected measurements using the individual electrodes in separate experiments are shown in Figure 13, while those obtained using the double-barrel arrangement are shown in Figure 14. Two operational amplifiers and two Ag/AgCl/KCl (3 M) reference electrodes were used in the simultaneous measurements. The distance between the antimony and the ISME tip was approximately 34 μm. The results of the simultaneous measurements were found to correlate well with the results of the individual measurements of pH and Zn^2+^ concentration (compare Figure 13 and Figure 14). In these line scan measurements, an area was found where the pH was strongly dependent on the Zn^2+^ concentration. The formation of zinc hydroxide released H^+^ ions; therefore, when the concentration of the zinc(II) ions was higher, the pH shifted slightly towards the acidic range. Furthermore, at large negative potential values, where the zinc concentration was smaller, the pH changed to alkaline values.

## 4. Conclusions

The highly sensitive way of evaluating degradation reactions and protection methods with chemical selectivity provided by scanning microelectrochemical methods has been reviewed with respect to the scanning vibrating electrode technique (SVET) and the scanning electrochemical microscope (SECM).

The applicability of these techniques was illustrated using a practical example of a metal/coating system, which consists of exposing the cut edges of coil-coated galvanized steel to an aqueous saline environment. Proper sample preparation allowed various combinations of coating layers to be exposed separately, demonstrating the effect of the components in improving the corrosion resistance of the material while also looking at the possibilities and limitations of microelectrochemical techniques.

The applicability of the SECM to study the cut edges of coated metals was expanded with the introduction of the use of ion-selective microelectrodes as tips, which makes the SECM work potentiometrically. Spatial resolution images of the electrochemical reactivity associated with the components of the studied material were selectively monitored both in situ and in real time.

The fabrication of multi-barreled micropipette-based probes to contain multiple ion-selective microelectrode systems has paved the way for simultaneous detection. In this contribution, the localized distributions of zinc(II) ions and pH arising from cut edges of coil-coated steel immersed in a 1 mM NaCl solution were shown.

A procedure was presented to obtain 2D images of a complete cut edge system from a small selection of potentiometric SECM line scans has been presented. Pseudo-maps were obtained when the step sizes on the *X* and *Y* axis are significantly different during the scanning procedure. These quasi-maps have been shown to provide satisfactory images of the spatial distributions and local changes occurring at the cut edges of coil-coated steel to follow their temporal evolution in situ.

## Figures and Tables

**Figure 1 molecules-27-02167-f001:**
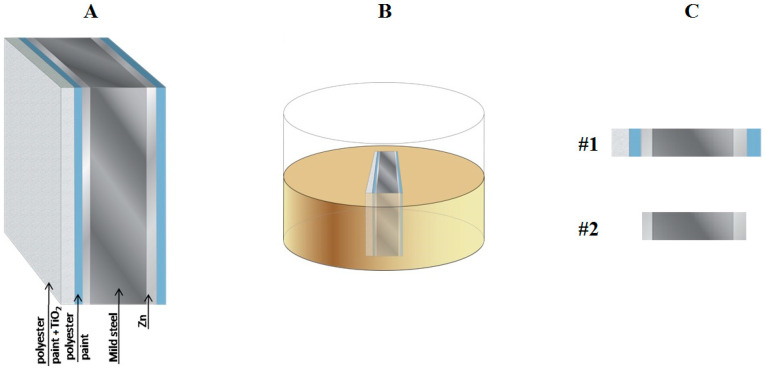
Sketches depicting the sample preparation process and samples characteristics: (**A**) view of the complete coil-coating cut edge system; (**B**) epoxy resin sleeve containing the complete coil-coating cut edge system; and (**C**) top view of the two different cut edge configurations resulting before and after removal of the polymer layers at either side.

**Figure 2 molecules-27-02167-f002:**
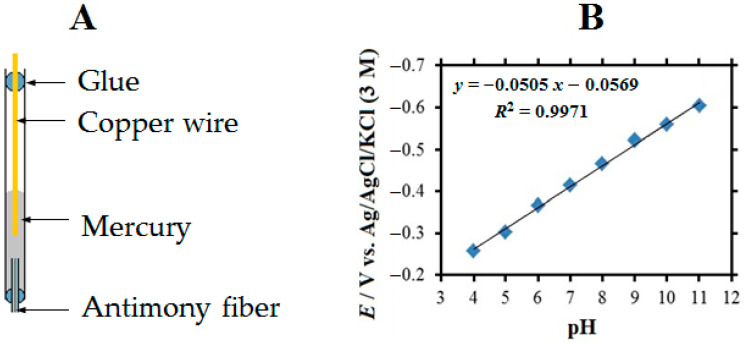
(**A**) Schematic and (**B**) calibration curve of a pH-sensitive single-barrel antimony microelectrode.

**Figure 3 molecules-27-02167-f003:**
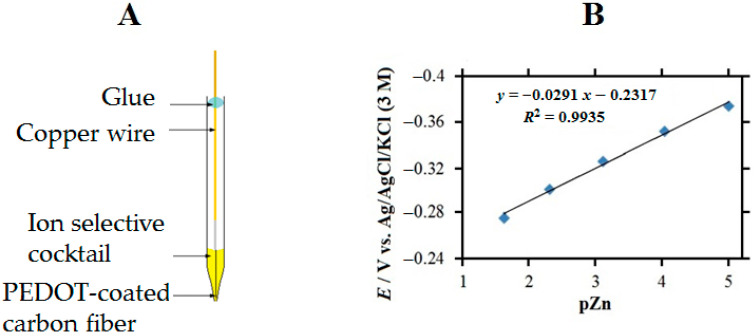
(**A**) Schematic and (**B**) calibration curve of a single-barrel Zn-ISME with internal solid contact.

**Figure 4 molecules-27-02167-f004:**
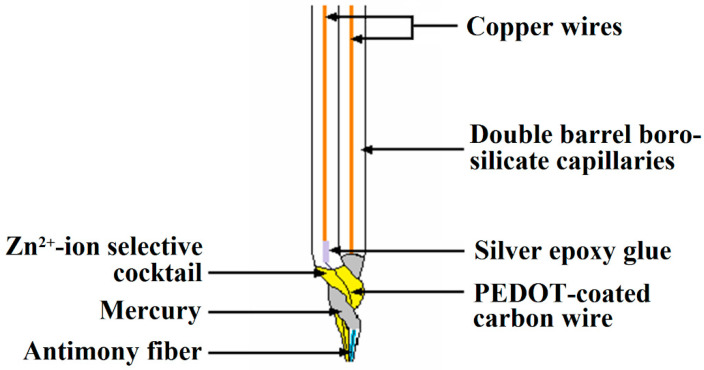
Schematic of the double-barrel Zn-ISME/Sb probe for combined potentiometric detection of Zn(II) ion concentration and pH distributions in the SECM.

**Figure 5 molecules-27-02167-f005:**
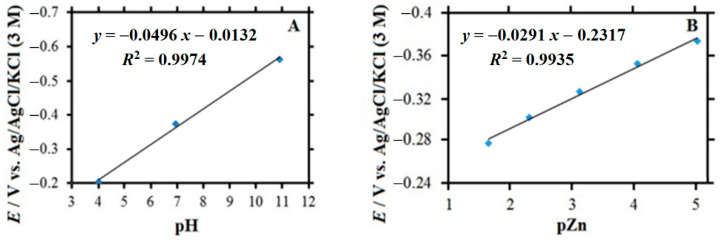
Calibration curves of the microelectrodes sensitive to pH and Zn^2+^ ions in the double-barrel assembly. (**A**) Linear pH response for the antimony microelectrode for 4 ≤ pH ≤ 11; (**B**) linear behavior of the Zn-ISME for 1.5 ≤ pZn ≤ 5.

**Figure 6 molecules-27-02167-f006:**
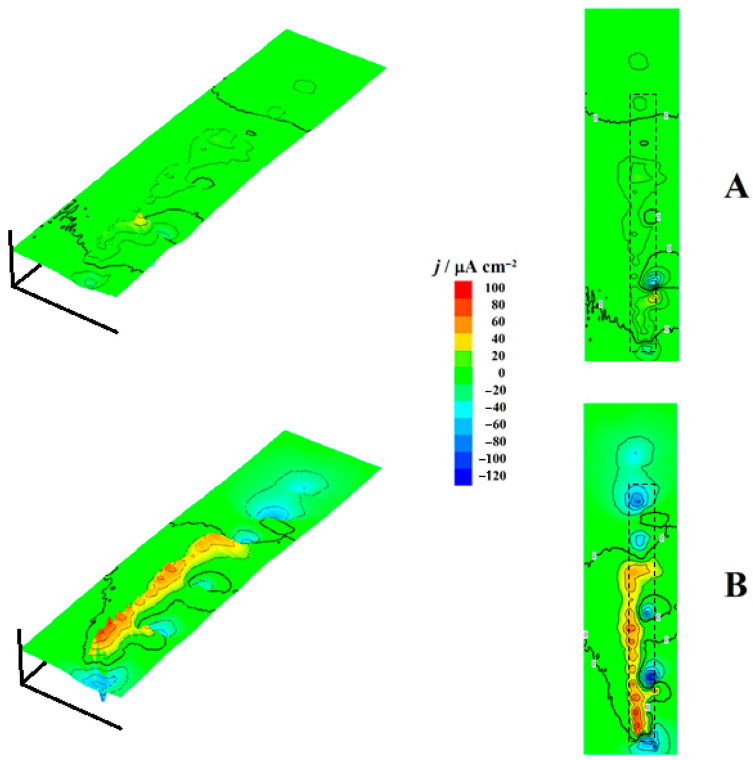
SVET maps recorded for the cut edge of a galvanized steel sample after removing the organic coatings at each side (i.e., corresponding to sketch #2 in Figure 1C). The images were recorded after (**A**) 15 and (**B**) 165 min immersion in 1 mM NaCl solution. The approximate location of the specimen is indicated by the drawn rectangles. The images represent, in *X* and *Y* directions, 1580 µm × 6000 µm. Values of *Z* axis: Ionic current, µA cm^–2^. Mean probe-substrate distance: 100 µm.

**Figure 7 molecules-27-02167-f007:**
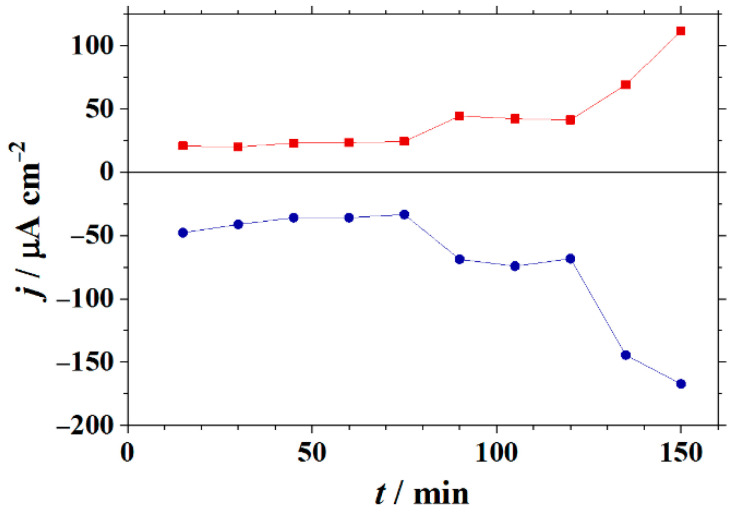
Time courses of current densities determined at the main peaks associated with adjacent cathodic (blue) and anodic (red) regions on SVET maps obtained over the cut edge of a galvanized steel sample after removing the organic coatings from each side (i.e., corresponding to sketch #2 in Figure 1C) immersed in 1 mM NaCl solution.

**Figure 8 molecules-27-02167-f008:**
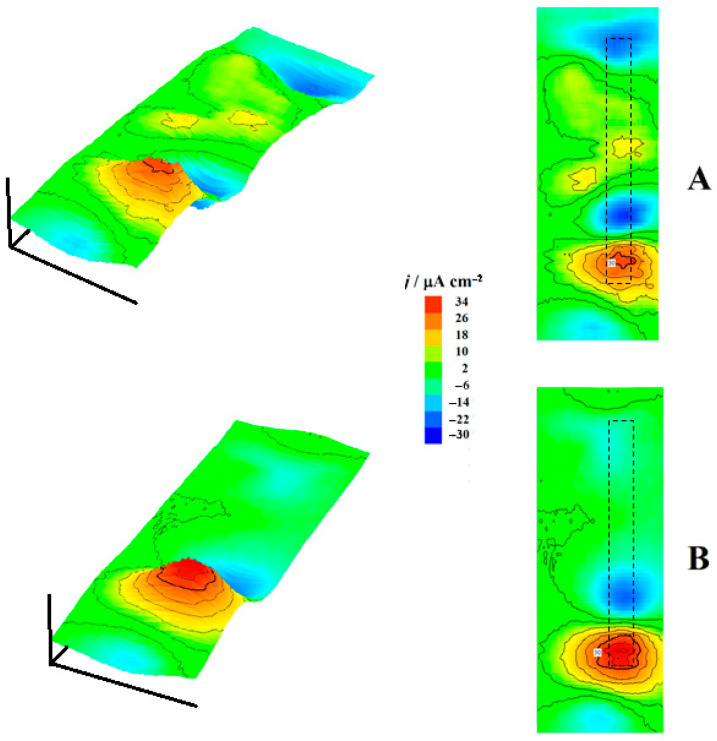
SVET maps recorded for the cut edge of a complete coil coating steel sample immersed in a 1 mM NaCl solution (corresponding to sketch #1 in Figure 1C). The images were obtained after (**A**) 15 and (**B**) 165 min immersion in the test solution. The approximate location of the specimen is indicated by the drawn rectangles. The images represent, in *X* and *Y* directions, 1010 µm × 2790 µm. Values of *Z* axis: Ionic current, µA cm^–2^. Mean probe-substrate distance: 100 µm.

**Figure 9 molecules-27-02167-f009:**
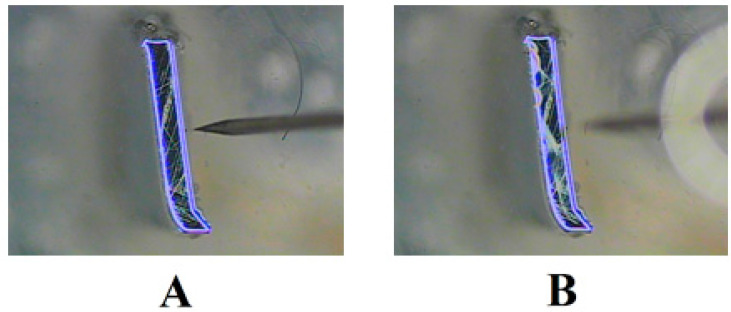
Optical micrographs of the cut edge of the complete coil coating steel sample immersed in a 1 mM NaCl solution. Duration of immersion in test solution: (**A**) freshly immersed, and (**B**) 165 min (immediately after SVET map recording in Figure 8B). The images were taken in situ using the video camera system coupled to the SVET equipment. The vibrating probe used to obtain the SVET maps can be seen in the center right of the micrographs.

**Figure 10 molecules-27-02167-f010:**
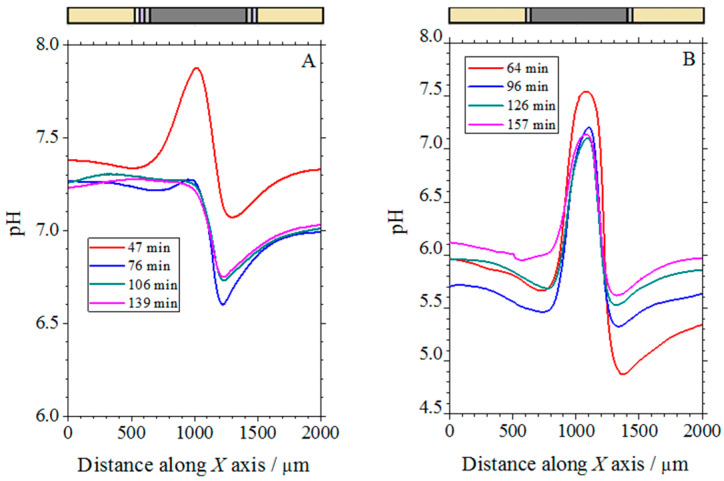
Line scans recorded with the pH sensitive antimony microelectrode above the cut edge of galvanized steel samples immersed in 1 mM NaCl solution. (**A**) Complete coil-coated system with organic layers of different thickness at each side (corresponding to sketch #1 in Figure 1C); (**B**) coil-coating system after removing the organic coatings at each side (corresponding to sketch #2 in Figure 1C). The lines were recorded from the left to the right.

**Figure 11 molecules-27-02167-f011:**
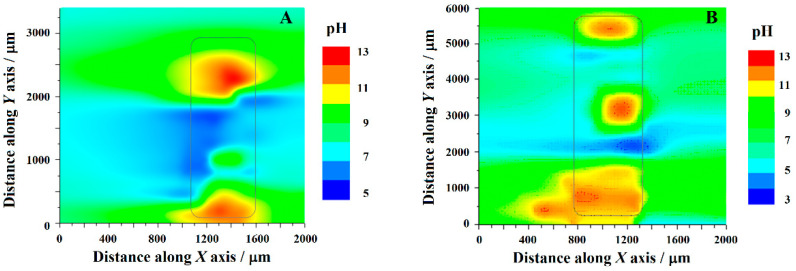
pH quasi-map of cut edges samples immersed in 1 mM NaCl solution. (**A**) Complete coil-coating system with organic coating layers of different thickness at each side (corresponding to sketch #1 in Figure 1C); and (**B**) coil-coating system after removing the organic coating layers at each side (corresponding to sketch #2 in Figure 1C). The approximate location of the specimen is indicated. The line scans were taken every 400 μm along the *Y* axis, and always from the right to the left.

**Figure 12 molecules-27-02167-f012:**
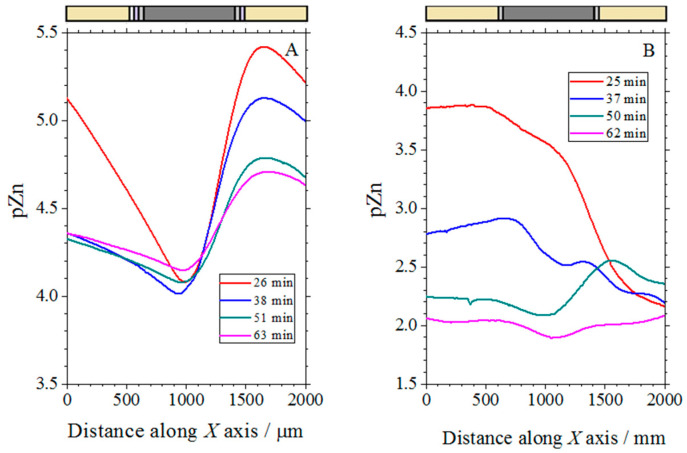
Line scans recorded with the Zn ISME above the cut edge of galvanized steel samples immersed in 1 mM NaCl solution. (**A**) Complete coil-coated system with organic layers of different thickness at each side (corresponding to sketch #1 in Figure 1C); (**B**) coil-coating system after removing the organic coatings at each side (corresponding to sketch #2 in Figure 1C). The lines were recorded from the left to the right.

**Figure 13 molecules-27-02167-f013:**
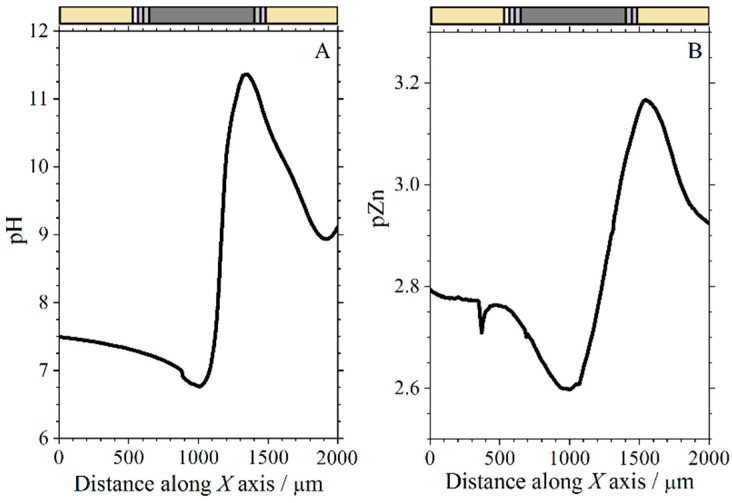
Line scans with pH-sensitive antimony electrode (**A**) and Zn^2+^-ISME (**B**) above the cut edge of painted galvanized steel (corresponding to sketch #1 in Figure 1C) immersed in 1 mM NaCl solution. The line scans were recorded in separate measurements.

**Figure 14 molecules-27-02167-f014:**
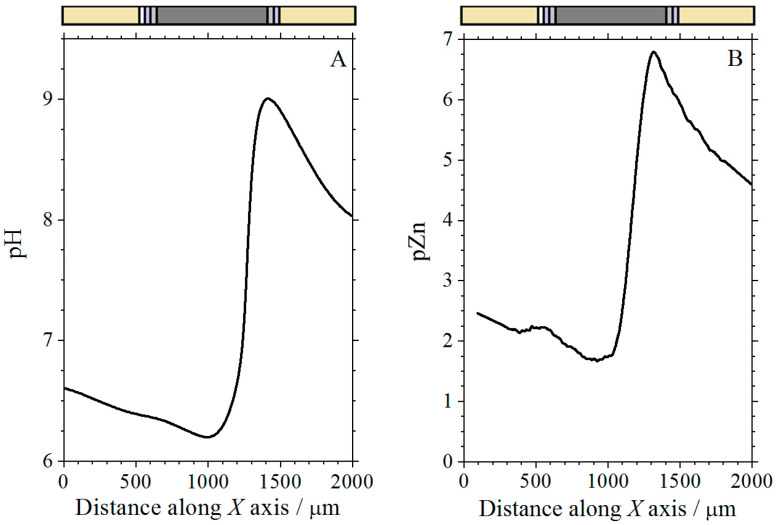
Simultaneous line scans recorded with a double-barrel microelectrode tip containing a pH-sensitive antimony electrode (**A**) and a Zn^2+^-ISME (**B**) above the painted cut edge of galvanized steel (corresponding to sketch #1 in Figure 1C) immersed in 1 mM NaCl solution.

## Data Availability

Not applicable.

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
