# Peer review of "Contributions of Microelectrochemical Scanning Techniques for the Efficient Detection of Localized Corrosion Processes at the Cut Edges of Polymer-Coated Galvanized Steel"

_molecules, 2022, doi:10.3390/molecules27072167_

Round 1

Reviewer 1 Report

In this study, spatially resolved information on corrosion reactions operating at the cut edges of coated metals was investigated by using microelectrochemical scanning techniques using a suitable selection of operation modes and scanning probes. This study achieves good originality but is insufficient for acceptance in as-received form. I think that the paper should be improved in these aspects:

  1. In abstract part, some obtained results should be included.
  2. The preparation method of mild steel coated with Zn in this study and their bonding strength should be provided.
  3. The Localized Electrochemical Impedance Spectroscopy (LEIS) is present in this study. Actually, Electrochemical Impedance Spectroscopy (EIS) is widely used to investigate the corrosion mechanism, for example, Xiaoting Shi, Yu Wang, Hongyu Li, et al, Corrosion resistance and biocompatibility of calcium-containing coatings developed in near-neutral solutions containing phytic acid and phosphoric acid on AZ31B alloy, Journal of Alloys and Compounds, 2020, 823, 153721; M. Soleymanibrojeni, H. W. Shi, I. I. Udoh, et al., Microcontainers with 3-amino-1,2,4-triazole-5-thiol for Enhancing Anticorrosion Waterborne Coatings for AA2024-T3, Progress in Organic Coatings, 137 (2019) 105336. The characteristics of EIS should be simply introduced.
  4. The level of English should be further improved. For example, “Unfortunately, the local potential and ionic current fluxes data obtained using SRET and SVET, as well as for impedance measurements, they (should delete “they”) lack chemical selectivity and thus can only detect the sites on the sample associated to either cathodic or anodic half cell reactions, but cannot serve to identify the chemical species involved in those reactions”.
  5. The equation of O2 + H2 + 4e- → 4OH- should be changed into O2 + 2H2O + 4e- → 4OH- or ……..

Author Response

In this study, spatially resolved information on corrosion reactions operating at the cut edges of coated metals was investigated by using microelectrochemical scanning techniques using a suitable selection of operation modes and scanning probes. This study achieves good originality but is insufficient for acceptance in as-received form. I think that the paper should be improved in these aspects:

In abstract part, some obtained results should be included.

Done. The following sentence has been included:

The ISME probes detected differences in the local concentrations of Zn2+ and OH- ions from the cut edges of a complete coil coating system compared to the same system after the polymeric layers were removed. In this way, it has been shown that the inhibitor loading in the polymer layers effectively contributes to reducing the corrosion rates at the cut edge, thus helping to extend the useful life of the sacrificial galvanized layer bonded directly to the steel matrix.

The preparation method of mild steel coated with Zn in this study and their bonding strength should be provided.

Since these materials were previously characterized using conventional electrochemical methodologies and more detailed descriptions were supplied there, the relevant references to those works were included as the new references 40 and 41.

The Localized Electrochemical Impedance Spectroscopy (LEIS) is present in this study. Actually, Electrochemical Impedance Spectroscopy (EIS) is widely used to investigate the corrosion mechanism, for example, Xiaoting Shi, Yu Wang, Hongyu Li, et al, Corrosion resistance and biocompatibility of calcium-containing coatings developed in near-neutral solutions containing phytic acid and phosphoric acid on AZ31B alloy, Journal of Alloys and Compounds, 2020, 823, 153721; M. Soleymanibrojeni, H. W. Shi, I. I. Udoh, et al., Microcontainers with 3-amino-1,2,4-triazole-5-thiol for Enhancing Anticorrosion Waterborne Coatings for AA2024-T3, Progress in Organic Coatings, 137 (2019) 105336. The characteristics of EIS should be simply introduced.

A mention to the surface-averaging technique EIS has been mentioned with a reference to a review describing its foundations and applications for characterizing corrosion protection by organic coatings which should help the interested reader (new reference [1]). Additionally, we thank the reviewer for pointing us to valuable references on the use of EIS that have been included in the manuscript as new references 2 and 3.

The level of English should be further improved. For example, “Unfortunately, the local potential and ionic current fluxes data obtained using SRET and SVET, as well as for impedance measurements, they (should delete “they”) lack chemical selectivity and thus can only detect the sites on the sample associated to either cathodic or anodic half cell reactions, but cannot serve to identify the chemical species involved in those reactions”.

Done accordingly.

The equation of O2 + H2 + 4e- → 4OH- should be changed into O2 + 2H2O + 4e- → 4OH- or ……..

The Reviewer is correct in pointing out this error which has been taken into account. It is the same equation that we have always used in related systems for the reduction of dissolved oxygen (cf. reference 42)

Reviewer 2 Report

The article is a follow-up of authors’ work on the applicability of scanning microelectrochemical methods in monitoring corrosion reactions at the microscale. In this work authors apply the scanning vibrating electrode technique (SVET) on a galvanized steel/coating system immersed in NaCl. The localized distributions of Zn(II) ions and pH arising from cut edges of coil coated steel were monitored using an ion-selective microelectrode (ISME) as a probe. The article conveys adequate insight into the field and the conclusions seem to be well supported by the experimental data. I think that the article merits publication in its present form. However, I am wondering if this article is suitable for the present journal. In any case I would recommend to the authors to make clearer the goal of the present study and its differentiation from previous similar studies.

Author Response

The article is a follow-up of authors’ work on the applicability of scanning microelectrochemical methods in monitoring corrosion reactions at the microscale. In this work authors apply the scanning vibrating electrode technique (SVET) on a galvanized steel/coating system immersed in NaCl. The localized distributions of Zn(II) ions and pH arising from cut edges of coil coated steel were monitored using an ion-selective microelectrode (ISME) as a probe. The article conveys adequate insight into the field and the conclusions seem to be well supported by the experimental data. I think that the article merits publication in its present form.

We thank the Reviewer for the support comments regarding our work.

However, I am wondering if this article is suitable for the present journal.

It must be noticed that our contribution is made in response to an invitation for a special issue of the journal MOLECULES on the topic "Electrochemistry and Corrosion Protection of Metallic Materials", and it should fit well in it.

In any case I would recommend to the authors to make clearer the goal of the present study and its differentiation from previous similar studies.

Agreed. The last paragraph of the Introduction section has been expanded to make clearer the goal of the present study:

“Additionally, procedures to obtain chemically-resolved images of the complete cut-edge exposed to the aggressive environment have been explored, allowing to overcome the previous limitation of SECM investigations that were based solely on a small selection of scan lines due to slower acquisition times compared to SVET or LEIS.”

Reviewer 3 Report

The manuscript presents and interesting research on corrosion detection method development but some improvement is needed.

-please provide some references on Zn2+ ion-selective microelectrode so that readers can understand the basic principle of how it works. Is this electrode sensibility influenced by solution pH?

- is it possible to show boundary between steel and coatings in images 6 and 8? It would be useful to know that (similar to that shown in Figure 10).

-line 366 – „ The distinctive protective effect of the organic coating and the galvanized layers was also distinguished, as the total pH change was less than one unit in the case of the complete coil coating system“ – in principle coatings present barrier effect, but only for the surface area they are covering not for the neighbouring surface. Do you believe that corrosion inhibitor incorporated in the coating is responsible for that (as it may dissolve from the coating and precipitate over steel)?

Author Response

The manuscript presents and interesting research on corrosion detection method development but some improvement is needed.

please provide some references on Zn2+ ion-selective microelectrode so that readers can understand the basic principle of how it works. Is this electrode sensibility influenced by solution pH?

Both the operation principle of the ISME (namely the measurement of a Donnan potential) and the pH influence on its response have been included in the manuscript together with relevant references for more detailed information (new references 35 and 42). The following additions have been made in the main text:

“This ISME selectively reaches an equilibrium Donnan potential that varies logarithmically with Zn2+ ion concentration as described elsewhere [26].”

“In a previous work [42], the pH dependence of the potential of this Zn2+-ISME was shown to be negligible for pH £ 9.”

is it possible to show boundary between steel and coatings in images 6 and 8? It would be useful to know that (similar to that shown in Figure 10).

Done. Boundaries for the approximate position of the cut edge in the maps have been included in Figures 6 and 8.

-line 366 – „ The distinctive protective effect of the organic coating and the galvanized layers was also distinguished, as the total pH change was less than one unit in the case of the complete coil coating system“ – in principle coatings present barrier effect, but only for the surface area they are covering not for the neighbouring surface. Do you believe that corrosion inhibitor incorporated in the coating is responsible for that (as it may dissolve from the coating and precipitate over steel)?

Fully agreed. The effect of inhibitor species contained in the polymer coating layer was not sufficiently well described in the manuscript, although left implicit in the previous version of the manuscript. In order to clarify this observation, the following sentence was included in the manuscript:

“Since the polymer coating layers cannot maintain the barrier effect at the cut edge because the metals both in the galvanized layer and in the steel plate are directly exposed to the electrolytic medium, the beneficial protection effect they provide must come from the action of the corrosion-inhibiting species dispersed in the polymer matrix, which are effectively released from the coating to precipitate on the bare metal surfaces.”